# Habitat Suitability Assessment of Black-Necked Crane (*Grus nigricollis*) in the Zoige Grassland Wetland Ecological Function Zone on the Eastern Tibetan Plateau

**Junjun Bai [1,2], Peng Hou [1,2,*], Diandian Jin [2], Jun Zhai [2], Yutiao Ma [2,3] and Jiajun Zhao [1,2]**

[1] Chinese Research Academy of Environmental Sciences, Beijing 100012, China; junj_bai@163.com (J.B.); zhaojiajun2345@163.com (J.Z.)
[2] Satellite Environment Application Center, Ministry of Ecology and Environment, Beijing 100094, China; jin_diandian@163.com (D.J.); zhaij@lreis.ac.cn (J.Z.); qwertbng@126.com (Y.M.)
[3] School of Geomatics and Spatial Information, Shandong University of Science and Technology, Qingdao 266000, China
* Correspondence: houpcy@163.com

**Abstract:** Habitat suitability assessment is critical for wildlife population conservation and management planning. The MaxEnt model is widely used in species habitat suitability modeling. In order to investigate the habitat status of the black-necked crane in the Zoige grassland wetland ecological function area, this study evaluated the habitat suitability of the black-necked crane using the MaxEnt model with 152 occurrence records and 13 environmental variables. Based on the Akaike information criterion corrected for the small sample size, the best optimal parameter combination (feature class LQPHT, regularization multiplier 3.0) was selected. The results show that the Maxent model had good accuracy with an area under the curve (AUC) value of 0.895. Distance to roads, average summer precipitation, distance to lakes, and altitude are the dominant environmental variables affecting the potential distribution of black-necked cranes, with the contribution rates of 31.6%, 29.8%, 14.3%, and 8.2%, respectively. The suitable habitat area of black-necked cranes is mainly distributed in and around the Zoige Wetland National Nature Reserve, accounting for 46.49% of the Zoige Grassland Wetland National Key Ecological Function Zone. The potential distribution area has a tendency to spread to Hongyuan County in the south of the functional zone, and the unsuitable habitat is mainly distributed in the high-altitude area in the southwest of the functional zone. This study recommends focusing on the distribution area of black-necked cranes around Zoige Wetland National Nature Reserve and Hongyuan County to improve conservation strategies and strengthen protection efforts.

**Keywords:** Zoige grassland wetland; black-necked crane; species distribution model; MaxEnt; habitat suitability assessment; model optimization

## 1. Introduction

Biodiversity refers to the diversity of all living things on earth and is the basis for human survival and development. Furthermore, biodiversity conservation can enhance human well-being. Habitat loss and fragmentation are the major cause of biodiversity loss [1]. Habitats, as places where species can live and reproduce, can provide for individuals, populations, and communities to complete their cycle. Habitat suitability assessment is a critical component of species conservation research and an important indication of habitat quality [2,3]. The assessment of the habitat suitability of wildlife has become a hot issue in the study of the changes in the global species spatial pattern [4–7]. Therefore, understanding the habitat conditions of the studied species, that is, exploring and analyzing the environmental variables affecting the distribution of species and identifying potential geographic distribution areas, is required in order to provide scientific theories for effective species conservation and protected area management planning [8].

Species distribution models are important tools for studying the habitat suitability of species, identifying potential species distribution areas, revealing possible influencing factors, and providing an important scientific basis for biodiversity conservation [9,10]. Currently, there is extensive scientific literature on species distribution models [11]. Based on whether species occurrence records are necessary when the model is created, habitat suitability models are classified into three categories: mechanistic models, statistical models, and niche models [7,10,12–14]. The commonly used model are ecological niche factor analysis (ENFA) [15], random forest (RF) [16], Maximum Entropy Models (MaxEnt) [9,17], generalized linear model (GLM) [18], generalized additive model (GAM) [18], and artificial neural networks (ANN) [19]. Among them, the MaxEnt model relies on species occurrence records and environmental variables and has broad applicability, allowing for less bias and more accurate results [20–22]. At the same time, it can still obtain better results when compared with other species distribution models and is widely used to assess the distribution of wildlife habitats [17,23].

The black-necked crane (*Grus nigricollis*) is listed as a national I-class protected animal by China and a near-threatened species (NT) by the International Union for Conservation of Nature [24,25]. Only black-necked cranes inhabit the plateau, which is mainly found on the Qinghai–Tibet Plateau and the Yunnan–Guizhou Plateau. So far, the total number of black-necked cranes around the globe has reached 10,000–10,200 [26]. The majority of the black-necked cranes overwinter in the low-altitude areas of the Qinghai–Tibet Plateau, the Yunnan–Guizhou Plateau, Bhutan, and southern Tibet, and breed in the Zoige wetland at the northeastern end of the Qinghai–Tibet Plateau [26]. With around 2600 black-necked cranes [27], the Zoige Wetland is the largest swamp wetland on the Eastern Tibetan Plateau and one of the most important nesting places for them [28]. The Zoige Wetland National Nature Reserve was established to protect local biodiversity, while the Zoige Grassland Wetland Ecological Function Zone was established to provide regional ecological security. Although the black-necked crane population is increasing, it is also threatened by the reduction in wetland area during the 21st century, as melting glaciers and permafrost degradation caused by local economic development, as well as future global climate change, may negatively affect shallow wetlands [29,30].

Researchers conducted surveys on the population abundance, distribution [31–33], and migration patterns of black-necked cranes [34,35]. In terms of the behavioral ecology of black-necked cranes, the feeding time during the overwintering period is mainly regulated by humidity indirectly [36,37], and the breeding season is mainly distributed in meadows and marsh meadows [38]. The study of Kong et al. suggested that the impact of predator threat and human disturbance on black-necked cranes should be considered in future tourism management, and a safe distance should be planned reasonably [39]. Human disturbance, food, and water conditions are the key environmental variables impacting the habitat quality of black-necked cranes in the Napa Sea wetland [40], according to studies on their habitat choices. Furthermore, land-use change influences the feeding and nocturnal habitat selection of black-necked cranes [41], and precipitation is another key factor impacting their habitat [42]. The black-necked cranes distributed in the Zoige Wetland are mainly influenced by altitude and autumn climate [43]. The distance from the cultivated land, the distance from the water, and the dominant vegetation are the main environmental factors affecting the distribution of black-necked cranes in the Caohai National Nature Reserve, Guizhou, China [44].

Previous studies have mostly focused on the migration routes, population changes, behavioral ecology, and habitat quality of black-necked cranes. At the spatial scale, the focus has been on the global distribution of black-necked cranes, especially in China, but most studies at the regional level have focused on the distribution, influencing factors, and the conservation status of the wintering areas in Yunnan and Guizhou, with insufficient attention to the breeding sites in Zoige, Sichuan [31–44]. The Zoige Wetland is the largest breeding place for black-necked cranes, and it is a key element of their life cycle. To conserve species and ecosystems, nature reserves are defined and zoned. Therefore, further

research on the distribution and habitat of black-necked cranes in breeding grounds is needed to narrow the gap with actual local conservation actions. Thus, in this study, we selected the Zoige Grassland Wetland Ecological Function Area, which is located in the core area of the Zoige wetland. The optimized maximum entropy model was used to predict the distribution of the black-necked cranes, analyze the main environmental factors affecting the distribution of black-necked cranes and their habitat distribution characteristics, and provide a scientific basis for the formulation of future measures for the efficient conservation and management of black-necked cranes.

## 2. Materials and Methods

### 2.1. Study Area

Zoige Grassland Wetland National Key Ecological Function Areas (Figure 1) (31°51′—34°18′ N, 101°6′—103°38′ N), located in the center of the Zoige Wetland, the largest marsh wetland on the Qinghai–Tibet Plateau, is an important part of the conservation land system of the Qinghai–Tibet Plateau. Meanwhile, the area is one of the world's most important alpine marsh wetlands, with a unique role in global climate change and regional ecological security [28,45]. There are about 2600 black-necked cranes in the entire Zoige wetland, which has the largest breeding population of black-necked cranes in the world. The region's unique geological, climatic, and hydrological natural conditions provide a favorable environment for black-necked crane survival and reproduction. The functional zone is at an altitude of 2442–4921 m, including Aba County, Zoige County, and Hongyuan County, with a total area of about 28,500 km². It is located in the watershed of the Yellow River and Yangtze River system, with abundant wetland peat resources, which play an important role in water conservation, hydrological regulation, and biodiversity maintenance of the Yellow River basin [27]. The National Main Functional Zone Plan, which defined 25 national key ecological functional zones, was promulgated and implemented in China in 2010 [46]. Among them, the Zoige Grassland Wetland National Key Ecological Function Zone is an important water conservation type zone in China, serving as a demonstration environment for people living in harmony with nature. The Zoige Wetland has an annual average temperature of 0.7–1.1 °C, with January temperatures of −10.5–7.9 °C, July temperatures of 10.9–11.4 °C, and annual average precipitation of 650–750 mm [47]. The Zoige wetland primarily protects rare wild species such as black-necked cranes, white storks (*Ciconia ciconia*), and the plateau swamp wetland habitat [48].

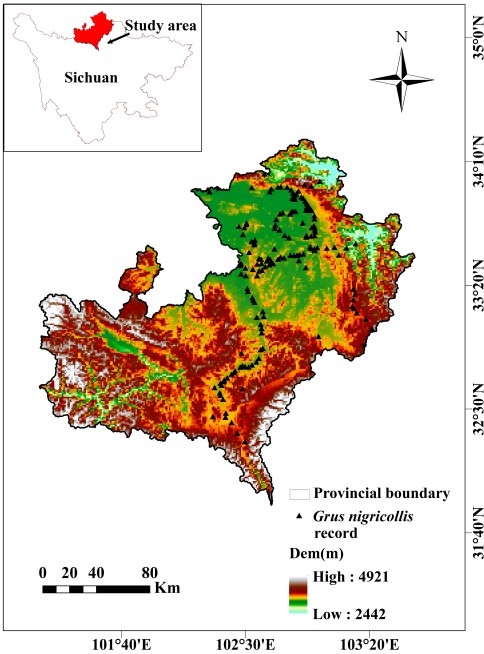

**Figure 1.** The presence data of black-necked crane.

## 2.2. Collection and Processing of Sample Data

Species distribution data were obtained from the published literature [49], field survey sites provided by Li et al. [50], and downloaded from the Global Species Diversity Information Database (http://www.gbif.org, accessed on 1 April 2022) and China Citizen Science Platform (http://www.birdreport.cn, accessed on 5 March 2022) [51,52], with record sites prior to 2010 removed. To eliminate duplicate, geographically and temporally unclear black-necked crane occurrence records and prevent covariance in environmental variables from affecting the model's accuracy, the SDM Toolbox was used, the GIS toolkit operating based on Python, version 2.5, developed by the Brown Lab et al., in Durham, America. Spatial filtering using the tool to ensure that only one point per grid (1 km × 1 km) was retained, resulting in 152 occurrence records (Figure 1, Table S1).

## 2.3. Screening and Determination of Environment Variables

Wildlife habitats must have the ability to provide their habitat, water, and food, so the distribution of wildlife is subject to a combination of climatic factors and human activities. Based on the principles of habitat suitability evaluation, combined with the previous studies analyzed above, the characteristics of the study area and the behavioral and ecological characteristics of black-necked cranes [36–38,40–44], the paper identified four major categories of factors related to the distribution of suitable habitats for black-necked cranes: climatic conditions, geomorphic types, foraging conditions, and human activities. First of all, in terms of climatic conditions, black-necked cranes return from migration in March and prefer to nest and breed near wetland marshes or in the center of shallow lakes, followed by a vital period of growth and development from June to August. The spring and summer precipitation resources are sufficient for vegetation growth as well as fish and shrimp spawning, which provides abundant food for the black-necked cranes. Considering the changes in behavioral habits of black-necked cranes in different seasons, factors such as average temperature and average precipitation from 2015 to 2020 were selected (bio1–bio10). The data were obtained from the Chinese 1 km resolution monthly precipitation dataset (1901–2020) of the National Tibetan Plateau Science Data Center [53]. Secondly, the geomorphic types (bio11–bio13) include elevation, slope, and aspect, and the data were downloaded from the Resource and Environment Science and Data Center (https://www.resdc.cn/, accessed on 4 March 2022). Thirdly, the foraging conditions (bio14–bio15, bio19) included the normalized difference vegetation index (NDVI), distance to rivers, and distance to lakes. The former data were obtained from the 2015–2020 NASA MODIS product data MODIS09A1 (http://ladsweb.nascom.nasa.gov/, accessed on 3 June 2021), and the latter two data were obtained from the China National Catalogue Service For Geographic Information (http://www.webmap.cn/, accessed on 22 January 2022). Finally, human activities (bio16–bio18) include the distance from roads, the distance from settlements, and land-use types. The first two data come from the National Geographic Information Resource Catalog Service System (http://www.webmap.cn/, accessed on 22 January 2022), and the latter data come from the Satellite Environment Application Center of the Ministry of Ecology and Environment of China. The environment variable details and sources are in Table 1.

**Table 1.** Variables used for modeling.

| Code | Environmental Variable | Source |
|------|------------------------|--------|
| Bio1 | Average spring precipitation | |
| Bio2 | Average summer precipitation | |
| Bio3 | Average autumn precipitation | |
| Bio4 | Average winter precipitation | |
| Bio5 | Average precipitation | |
| Bio6 | Average spring temperature | |
| Bio7 | Average summer temperature | |
| Bio8 | Average autumn temperature | |

**Table 1.** *Cont.*

| Code | Environmental Variable | Source |
|------|------------------------|--------|
| Bio9 | Average winter temperature | |
| Bio10 | Average temperature | http://data.tpdc.ac.cn/, accessed on 2 March 2022 (Bio1-Bio10) |
| Bio11 | Aspect (°) | |
| Bio12 | Altitude (m) | |
| Bio13 | Slope (°) | https://www.resdc.cn/, accessed on 4 March 2022 (Bio11-Bio13) |
| Bio14 | Distance to lakes (m) | |
| Bio15 | Distance to rivers (m) | |
| Bio16 | Distance to roads (m) | |
| Bio17 | Distance to settlements (m) | http://www.webmap.cn/, accessed on 22 January 2022 (Bio14-Bio17) |
| Bio18 | Land use | http://www.secmep.cn/, accessed on 11 August 2021 |
| Bio19 | Normalized difference vegetation index | http://ladsweb.nascom.nasa.gov/, accessed on 3 June 2021 |

Applying all of the environmental factors to the model modeling would result in overfitting due to the possible correlation between them. Therefore, the study used the ENMTools, version 1.0.6, developed by Warren et al., an R package for correlation analysis of each environmental factor, which does not depend on the distribution data and is able to obtain reliable results [54]. Figure 2 shows the environmental factor correlation heat map after processing the correlation plot with the corrplot, an R package, version 0.92, developed by Wei et al.; the minor environmental variables with $|R| \geq 0.9$ between the two environmental factors were excluded. Finally, the Jackknife method was used to screen again to remove the environmental factors with zero contribution rate, and only 13 environmental factors (bio1–bio4, bio10–bio16, bio18–bio19) were retained.

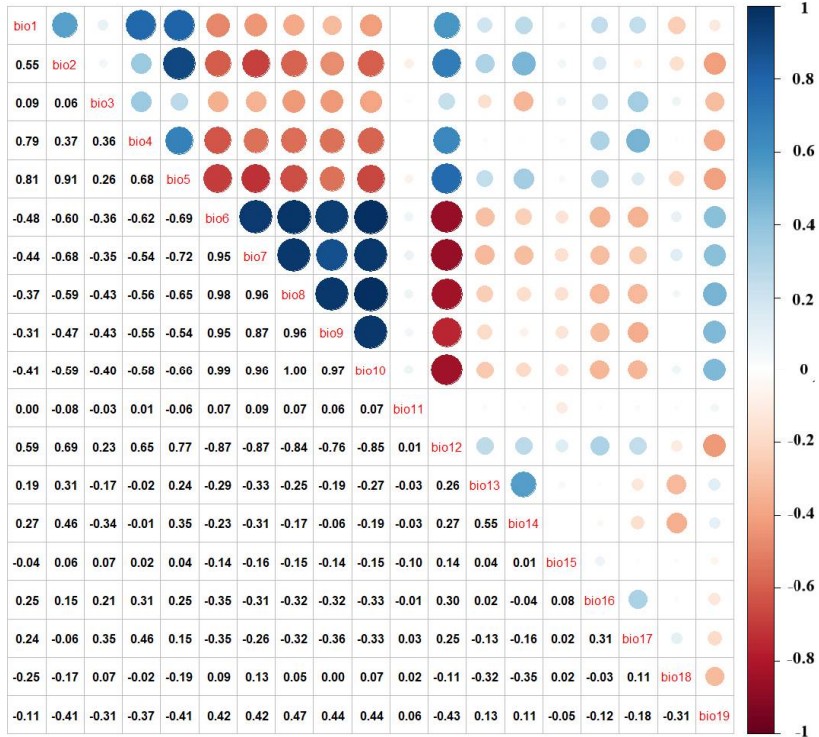

**Figure 2.** Heat map for correlation analysis of environmental factors. The darker the blue and red circles, the greater the correlation between the two environmental factors.

### 2.4. MaxEnt Model Optimization and Selection

The MaxEnt model has strong applicability and can use the area enclosed by the receiver operating characteristic curve (ROC) and the abscissa, that is, the AUC value, to evaluate the model prediction accuracy [9,20–22]. Moreover, the MaxEnt model's predictions based on default parameters are conservative, and the complexity of MaxEnt can be changed by altering the model parameter settings to predict the potential distribution of species more reasonably [7,55]. The study utilized the Enmeval, version 2.0.3, developed by Muscarella et al., an R package in R (v4.0.5) to optimize the MaxEnt model [56], which contains feature combination (FC) and regularization multiplier (RM). The feature class included Linear features (L), Quadratic features (Q), Product features (P), Hinge features (H), and Threshold features (T). The regularization multipliers were set to 1–4 with a 1 interval each, and the six feature combinations offered by the MaxEnt model (L, H, LQ, LQH, LQHP, and LQHPT) were merged to generate 24 combinations. The Akaike information criterion corrected for small sample size (AICc) was used as an indicator to determine the RM and FC of the model [57].

The black-necked crane occurrence records and the above 13 environmental factors were imported into the MaxEnt model, and the other settings were as follows: RM and FC values under the optimal parameters were input, 10-fold cross-validation was selected, the number of repetitions was 10, and the Jackknife method was chosen to test the importance of each environmental factor, and the output results were Logistic format. The model prediction results were examined using the area AUC under the ROC curve, and the value of AUC was taken in the range of 0–1, and the closer the value was to 1, the higher the model prediction accuracy. The AUC values are 0.5–0.6, unqualified; 0.6–0.7, poor; 0.7–0.8, fair; 0.8–0.9, good; and 0.9–1.0, excellent.

### 2.5. Habitat Suitability Classification of Black-Necked Cranes

Species predictive distribution maps show species preferences for habitat as probabilities (0–1), with the closer the value to 1, the more suitable the species distribution. The selection of thresholds generally follows three principles: objectivity, equivalence, and discriminative power [58]. The threshold is generally determined based on the omission error or based on the sensitivity and specificity of the prediction results. The former does not consider commission error, while the latter comprehensively considers omission error and commission error. The model maximum training sensitivity plus specificity (MTSS) belongs to the latter and satisfies the three principles of threshold selection [28]. MTSS and balance training omission and predicted area and threshold value (TPT) were selected as classification thresholds for suitable and low suitable habitats, respectively, to reclassify the MaxEnt model outputs into unsuitable, low suitable, moderately suitable, and highly suitable habitats [58–60]. Finally, the Reclassify tool of ArcGIS software, version 10.8, developed by Environmental Systems Research Institute, in RedLands, America, was used to count and calculate the area of the corresponding distribution area for each class.

## 3. Results and Analysis

### 3.1. MaxEnt Optimal Model and Accuracy Evaluation

Based on 152 occurrence records and 13 environmental factors, this study used the Enmeval package to invoke MaxEnt to predict the potential distribution area of black-necked cranes. The model with the lowest AICc value (i.e., ΔAICc = 0) is considered the best model out of the current suite of models [55,56]. When the model was the default parameter, FC = LQHPT, RM = 1, ΔAICc = 296.00, and when the model parameter was set to FC = LQHPT, RM = 4, ΔAICc = 0 (Table 2), the AICc value was the smallest and the model with this parameter was the optimal model. Figure 3 shows the results of comparing different parameters in the model. When the model parameters are set to FC = LQHPT, RM = 4, compared with the default parameters, Mean.AUC is nearly the same (decreased by 4.00%), but the difference between the AUC Values (Auc.diff.avg) decreased by 45.56% and 10%, and the training omission rate (OR10) decreased by 60.07%, the latter two are

lower than the default values, indicating that the optimized model reduced overfitting, so FC = LQHPT, RM = 4 was set as the modeling parameter. The model was reconstructed using the optimized parameters to simulate the suitable area for black-necked cranes, and the model was repeated 10 times, obtaining a mean value of 0.895 for the test AUC (Figure 4), indicating that the prediction accuracy of the MaxEnt model reached a good level.

**Table 2.** Evaluation metrics of MaxEnt model generated by Enmeval.

| Type | Feature Combination | Regularization Multiplier | ΔAICc | Avg.diff.avg |
|---|---|---|---|---|
| Default | LQPHT | 1 | 296.00 | 0.0413 |
| Optimized | LQPHT | 4 | 0 | 0.0225 |

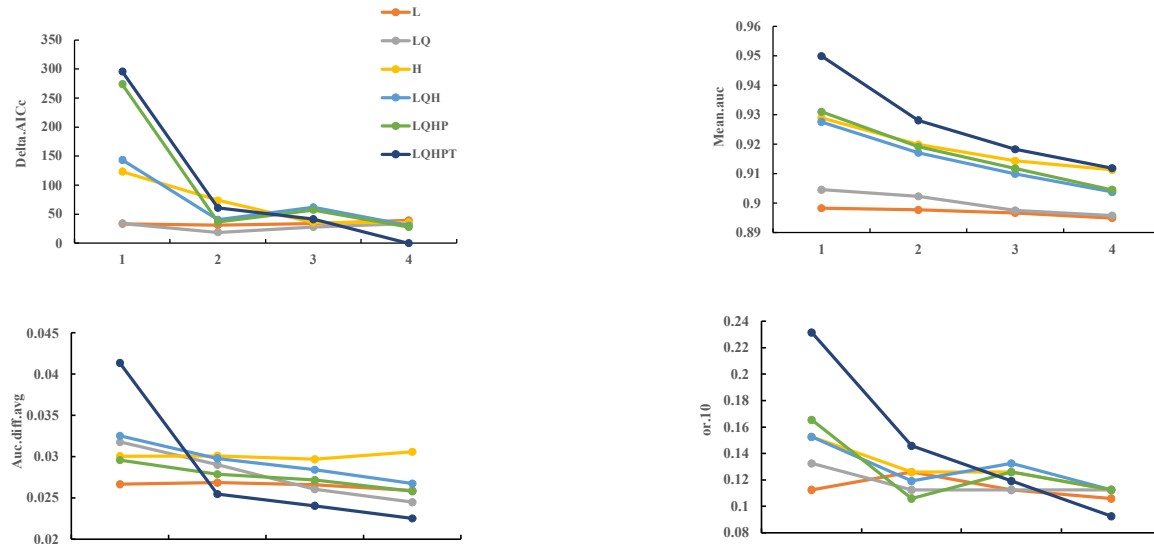

**Figure 3.** Performances of the maximum entropy model under different settings.

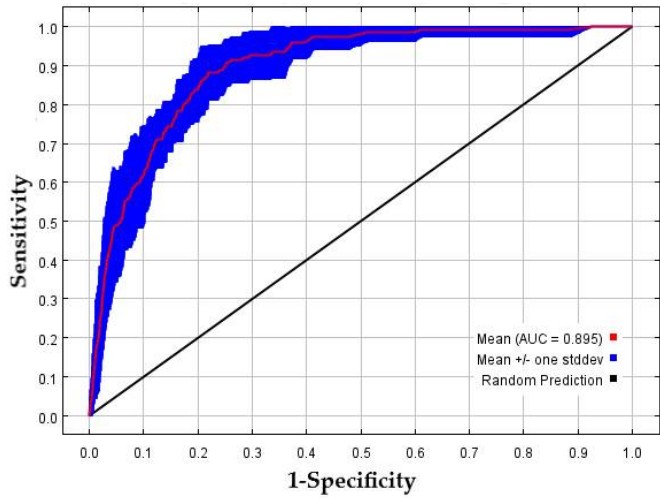

**Figure 4.** Performances of MaxEnt model under different settings.

*3.2. The Influence of Environmental Factors on the Distribution of Black-Necked Cranes*

The research used the Jackknife method to analyze the importance of 13 environmental factors affecting the habitat selection of black-necked cranes (Figure 5). The results of the contribution of environmental variables showed (Table 3) that distance to roads, average summer precipitation, distance to lakes, and elevation may be the major environmental factors affecting black-necked cranes, where the contribution rates were 31.6%, 29.8%, 14.3%,

and 8.2%, respectively, with a cumulative contribution rate of 83.9%; the secondary variables affecting the distribution of black-necked cranes were average summer precipitation, slope direction, NDVI, and the contribution rates were 5.6%, 2.5%, 2.3%, and 1.9%, respectively; and the percentages of other environmental factors were around 1%, indicating that the influence on the habitat suitability of black-necked cranes was small.

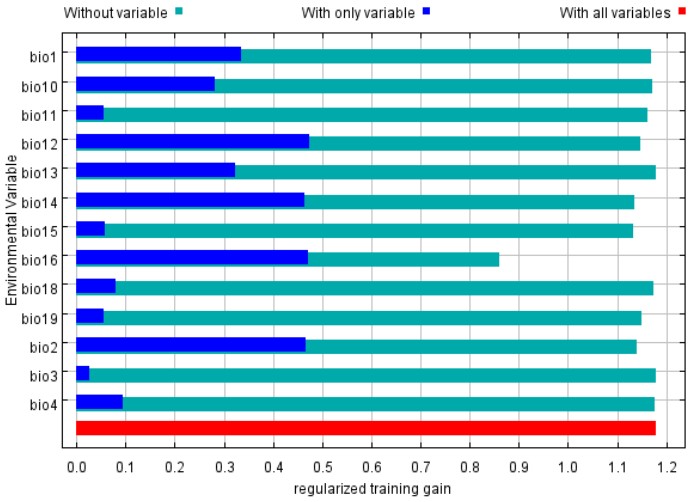

**Figure 5.** The importance of environmental variables evaluated by Jackknife testing.

**Table 3.** Contributions of the environmental variables to the MaxEnt model.

| Variable | Contribution/% | Variable | Contribution/% |
|---|---|---|---|
| Distance to roads (m) | 31.6 | Distance to rivers (m) | 1.9 |
| Average summer precipitation | 29.8 | Average winter precipitation | 1.1 |
| Distance to lakes (m) | 14.3 | Slope (°) | 1.1 |
| Altitude (m) | 8.2 | Average autumn precipitation | 0.8 |
| Average spring precipitation | 5.6 | Land use | 0.4 |
| Aspect (°) | 2.5 | Average temperature | 0.3 |
| Normalized difference vegetation index | 2.3 | | |

### 3.3. Habitat Suitability Distribution of Black-Necked Cranes in Zoige Grassland Wetland Ecological Function Zone

In the MaxEnt model results, MTSS = 0.304 and TPT = 0.0623, so the thresholds for classifying more suitable and less suitable habitats for black-necked cranes are 0.304 and 0.0623, that is. 1–0.5 is the highly suitable habitat, 0.5–0.304 is the moderately suitable habitat, 0.304–0.0623 is the low suitable habitat, and 0.0623–0 is an unsuitable habitat. To obtain the suitable habitat distribution map of black-necked cranes in the functional zone (Figure 6), the above thresholds were applied to reclassify the model outputs into different habitat classes, and the area of each suitable distribution area was calculated separately. The statistics reveal that the highly suitable habitat for black-necked cranes in the Zoige grassland wetland ecological function zone is about 2356.17 km$^2$, accounting for 8.27% of the total function area, mainly in the Zoige National Nature Reserve in the northern part of the function zone. Low suitable habitat covers approximately 7899.43 km$^2$, accounting for 27.72% of the total functional zone, mainly in Zoige country and Hongyuan country; unsuitable habitat covers about 0.98 km$^2$, accounting for 53.51% of the total functional zone, mainly in Aba County, a high-altitude area in the southwest. Moreover, the study indicated that the highly suitable habitat for black-necked cranes in the Zoige National Nature Reserve was nearly 668.70 km$^2$, accounting for 39.22% of the whole nature reserve area and 28.38% of the highly suitable habitat area in the total functional zone. The highly suitable habitat is mainly in the Zoige National Nature Reserve, which is primarily composed of marshes, wetlands, grasslands, lakes, and rivers. The reserve has abundant summer

rainfall and is covered with small lakes, creating good foraging conditions for black-necked cranes. Black-necked cranes prefer to be distributed close to water sources, which can provide good nesting conditions. The moderately suitable habitat was approximately 475.04 km$^2$, accounting for 27.86% of the whole nature reserve area and 15.87% of the moderately suitable habitat area in the total functional zone. The moderately suitable habitat was distributed in and around the nature reserve area, where there are undulating hills that cause differences in precipitation and temperature. All suitable habitats for black-necked cranes took up 97.48% of the whole nature reserve area and 46.49% of the total functional zone. In contrast to the nature reserve, the land use types of Aba County are mostly woodland and grassland in the southwest of the functional zone, which does not meet the demands of black-necked cranes for foraging and breeding. The spatial distribution of environmental variables affecting the distribution of black-necked cranes is more concentrated in the nature reserve but is not consistent in the entire functional zone. Therefore, the suitable distribution area of black-necked cranes is largely distributed in the nature reserve area.

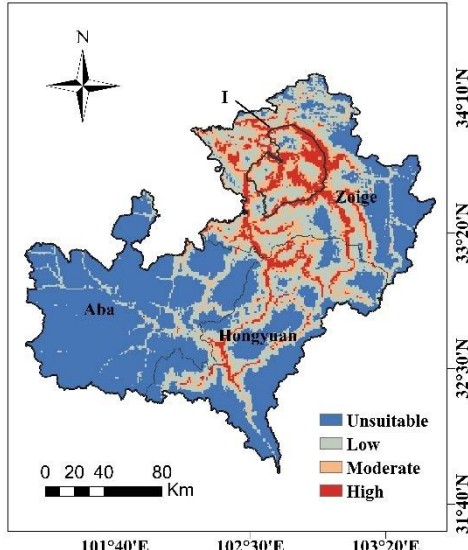

**Figure 6.** Map of the habitat suitability of black-necked crane in the Zoige Grassland Wetland National Key Ecological Function Zone (I. Zoige Wetland National Nature Reserve).

## 4. Discussion

MaxEnt model optimization generally includes correction of sampling bias, selection of environmental factors closely related to species distribution, and the optimization of model parameters [55,59,61–63]. Firstly, for the correction of sampling bias, the spatial filtering method used in this research ensures only one occurrence record in every 1 km × 1 km grid to reduce the overfitting phenomenon existing in model modeling [22], but this method may also overlook the ecological value of high-density areas of species distribution.

Second, habitat selection determinants for black-necked cranes may differ by region; thus, we recommend choosing environmental factors that are closely related to the species' distribution. Previous studies on the habitat selection of black-necked cranes show that human disturbance, food, and water conditions are the significant environmental variables determining the habitat quality of black-necked cranes [40]. Black-necked cranes distributed in the Zoige wetlands are mainly affected by altitude and autumn climate [43]. Distance to cultivated land, distance to water, and dominant vegetation are the impacting environmental factors affecting the distribution of black-necked cranes in the Caohai National Nature Reserve [44]. As a result, the research on black-necked crane habitat selection should be adapted to the study area's conservation situation and needs. After correlation and Jackknife analysis, only 13 environmental components were maintained in this study to reduce model overfitting. The results of the MaxEnt model prediction showed that the

habitat distribution of black-necked cranes was mainly influenced by distance to roads, summer precipitation, distance to lakes, and elevation. The distance to roads is closely related to the distribution of black-necked cranes, and by observing the occurrence records of black-necked cranes, their collection routes may be mostly distributed along the road, resulting in a strong contribution of the distance to roads factor, so the prediction results may be biased. The distribution of black-necked cranes is also mainly affected by spring precipitation, summer precipitation, and the distance to lakes, indicating that water and food are considered to be the main environmental factors, which is consistent with the results of previous investigations [40,42]. Black-necked cranes return from migration in March, preferring to nest and breed near wetlands and swamps or in the center of shallow lakes. After that, June-August is a critical period for the growth and development of black-necked cranes. The precipitation resources in spring and summer just provide sufficient conditions for vegetation growth and fish and shrimp reproduction and provide a rich food source for black-necked cranes [64]. However, the distance to cultivated land was discovered to be the primary determinant for the distribution of black-necked cranes in Dashanbao, Yunnan, and Caohai, Guizhou, China, because it makes up a significant portion of these research areas and offers more food than other environmental factors [41,44]. The majority of the swamp and lakes in our study region can offer the optimal environment for black-necked cranes to breed. From previous studies on the habitat of black-necked cranes, it can also be found that for other similar waterfowl, such as red-crowned cranes, Cao et al. used the MaxEnt model to reveal that the distance to roads, fishponds, and smooth cordgrass (*Spartina alterniflora*), and the distance to reed shoals and seepweed shoals, as well, were the main factors that influenced the selection of a wintering habitat by red-crowned cranes in the Yancheng Nature Reserve [65]. Na et al. found that habitat composition, water depth, and distance to roads and ditches were the most important habitat features for Red-crowned cranes in the Zhalong National Nature Reserve during the breeding season [66]. This also demonstrates that water and food are considered to be the main environmental factors in the distribution of waterfowl species. Black-necked cranes are the only cranes that live on plateaus among the 15 species of cranes in the world, mainly distributed in the Qinghai–Tibet Plateau and the Yunnan–Guizhou Plateau [26]. Therefore, compared with other cranes, Black-necked cranes have evolved good physiological adaptation characteristics and perfectly adapted to the living environment of high altitude and low temperature, which can explain that altitude is the main environmental factor for black-necked cranes distribution. In reality, the Zoige National Nature Reserve, the core area of the high suitable distribution of black-necked cranes, is strictly controlled, while the functional zone outside the nature reserve is a bit weaker and more vulnerable to human activities. For a more targeted analysis of the suitable distribution of black-necked cranes, it might be useful to further distinguish in detail the environmental variables affecting the distribution of black-necked cranes in the nature reserve and in the functional zone outside of the nature reserve, i.e., to select environmental factors separately for the nature reserve and functional zone, and then to superimpose these two results. In the future, we also need to supplement black-necked crane occurrence records to reduce data sampling bias and simulate the real black-necked crane distribution to provide a more scientific and theoretical basis for the nature reserve and functional zone planning and management.

Finally, in order to make the model findings more ecologically interpretable, a balance of model complexity is required for optimizing the model parameters [22]. Adjusting model complexity through FC and RM settings is a hot area in MaxEnt model research. With different sample sizes and feature combinations, the results of FC selection will be reflected in response curve plotting, making simple linear correlations or complex nonlinear correlations between environmental variables and distribution, which can have an impact on model fitting and prediction. Generally, simple models are simpler to understand ecologically, but if too few features are selected, such as selecting only the L function, as the sample size increases, the sampling bias will increase, resulting in a lower AUC value [67]. It is also argued that FC has little effect on the predictive ability of the model and that a

complex model will only slightly increase the AUC value [63]. The setting of RM is in order to balance the model fitting degree and extrapolation ability. When the RM value is set too low, the model is more likely to overfit and raise the omission error; when the RM value is set too high, the model becomes smooth, increases the misjudgment error, and loses its ability to discriminate in unsuitable areas. In this paper, the AUC values after model optimization are approximate to the default case, but the response curves obtained for some environmental factors are not particularly flat. Moreover, the increase in RM expands the error boundary range compared to the default setting. In practice, the selection of both FC and RM needs further judgment, and more studies are needed to show the relationship between the model parameter optimization results, sample size, and study subjects.

## 5. Conclusions

This study evaluated the habitat suitability of black-necked cranes in the Zoige grassland wetland ecological function area based on the optimized MaxEnt model and investigated the key environmental factors and suitable ranges affecting their distribution. The complexity and overfitting of the optimized model were minimized compared to the default parameters by optimizing two MaxEnt model parameters: FC and RM, establishing 24 combinations, and utilizing Akaike's small sample corrected information criterion (AICc) as an indication. The AUC value was 0.895, and the prediction results reached a good level. MaxEnt model prediction results show that the habitat distribution of black-necked cranes may be mainly influenced by distance to roads, summer precipitation, distance to lakes, and elevation. The suitable habitat area for black-necked cranes accounts for 46.49% of the total functional zone; with the suitable distribution area for black-necked cranes mainly located in the Zoige National Nature Reserve in the functional zone's north, the potential distribution area tends to spread to Hongyuan County in the functional zone's south, and the unsuitable habitat is mainly distributed in the high-altitude area in the southwest of the functional zone. The core distribution area is located in Zoige National Nature Reserve, so the delineation of the nature reserve can better protect black-necked cranes to some extent. The results of this study indicate that the summer precipitation and the spatial distribution of water resources have a major impact on black-necked crane distribution. It is recommended to strengthen the management of river and lake shorelines, protect the water ecological environment, actively communicate with local residents to raise their awareness of protection, and reasonably formulate the summer grazing intensity standards for grasslands to ensure the energy required by the black-necked cranes during the breeding and growth periods. In view of the impact of future climate change on wetland ecosystems and biodiversity, it is also necessary to strengthen the assessment and construction of protected area management capacity and to design evaluation indicators and management effectiveness assessment tools [68]. It is suggested to regularly monitor the population size of black-necked cranes and other wildlife, changes in the wetland area and human disturbance, assess changes in habitat quality of black-necked cranes, and adjust management activities in time to achieve conservation goals. At the same time, the contradiction between protection and development is still prominent, so the study recommends that attention be paid to the suitable distribution range of black-necked cranes on the vulnerable edge of the Zoige National Nature Reserve and Hongyuan County, as well as the impact of human activities on the distribution of black-necked cranes in these areas.

**Supplementary Materials:** The following supporting information can be downloaded at: https://www.mdpi.com/article/10.3390/d14070579/s1, Table S1: Occurrence records used for MaxEnt Model of black-necked crane in Zoige Grassland Wetland Ecological Function Zone.

**Author Contributions:** P.H., D.J., J.Z. (Jun Zhai), Y.M., J.Z. (Jiajun Zhao) and J.B. all contributed to the data analyses; J.B. performed analyses and led the writing; P.H. assisted with collect and check the data, and provided valuable comments in the paper writing. All authors have read and agreed to the published version of the manuscript.

**Funding:** This research was funded by National Key R&D Program of China (grant number 2021YFF0703903) and Major Projects of High-Resolution Earth Observation Systems of National Science and Technology (grant number 05-Y30B01-9001-19/20-4).

**Institutional Review Board Statement:** Not applicable.

**Data Availability Statement:** All occurrence records and environmental variables used in the manuscript are already publicly accessible, and we provided the download address in the manuscript.

**Acknowledgments:** Thanks to the Zoige National Nature Reserve Administration of Sichuan Province for providing data support.

**Conflicts of Interest:** The authors declare no conflict of interest.

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
