# Peer review of "Habitat Suitability Assessment of Black-Necked Crane (Grus nigricollis) in the Zoige Grassland Wetland Ecological Function Zone on the Eastern Tibetan Plateau"

_diversity, doi:10.3390/d14070579_

Round 1

Reviewer 1 Report

This manuscript focuses on an important topic such as modeling suitable habitat for a rare species of the spreading Black-necked Crane (Grus nigricollis) that inhabits in the Zoige Grassland Wetland Ecological Function Zone on the Eastern Tibetan Plateau.

It is necessary to know the most valuable sites in terms of its nesting and feeding to manage the population of this rare species,. This problem is solved by modeling based on registration data of this species. The model obtained by the authors has certain limitations, which are related to the fact that all observations of this species were mainly from the roads. More inaccessible areas were not surveyed or surveyed less frequently. However, the authors themselves recognize these limitations.

I think that the authors coped with their task in general and created a map of the most suitable habitats for the crane. The map will be useful in the management of the protection of this species.

I recommend this manuscript for publication after minor revisions.

I recommend correcting some technical typos and inaccuracies
Page 2, lines 95-98: here I would like to add references to the publications of these studies.
Page 2, line 96: "At" changed to "at"
Page 4, line 144: no link to sources in the Reference section of the gbif.org database
Page 12, line 410: article title missing

Reviewer 2 Report

This manuscript addresses the Habitat Suitability Assessment of a near threatened crane, with interesting implications to conservation that species. However, several methodological issues seriously question the results of the work.

First, as the authors indicate at the beginning of the discussion, the spatial filtering method used in this research ensures only one occurrence record in every 1 km ×1 km grid, but this method may also overlook the ecological value of high-density areas of species distribution. This is a serious problem, since it can significantly bias the final model, minimizing the importance of key areas repeatedly used by this species.

Second, the final model obtained could be biased by the sample data used. Specifically, the effect of roads seems to be conditioned by the data used to build the model.   In the discussion the authors indicate that the collection routes may be mostly distributed along the road, resulting in a strong contribution of the distance to roads factor, so the prediction results may be biased. In addition, I must indicate that the direction of the effect of each of the selected environmental variables is not indicated, which makes it extremely difficult to adequately interpret the results obtained.

Third, the discussion needs to be substantially improved, since it does not address the analysis of the model obtained in the light of similar models obtained in other waterfowl species.

Specific comments

Line 38. I suggest substituting the term “creature” for another, for example “species”.

Line 51. Obviously, there is an extensive scientific literature on species distribution models. There are many more than 25 species distribution models. I suggest rewriting this sentence or deleting it.

Line 62. It is obvious that if the scientific name of the species is Grus nigricollis, the genus is Grus. I suggest shortening this sentence by deleting "belongs to the Crane genus of the Crane family".

Line 64. I suggest replacing “World Conservation Union” with “International Union for Conservation of Nature”.

Line 96. Replace "behavioral ecology, and habitat quality of black-necked cranes; At the spatial scale …" with "behavioral ecology, and habitat quality of black-necked cranes. At the spatial scale …".

Line 133. I suggest substituting the term “creature” for another, for example “species”.

Line 137. Figure 1 needs to be improved. The upper left box does not identify the geographical area. The legend of this figure must be completed so that the figure as a whole is self-explanatory.

Line 178. Table 1 is incorrect. It is a table located in the Material and methods section and it does not make sense to put in it a result referring to the contribution of each environmental variable in the final model.

Lines 185-186. The explanation referring to the colors used to identify the correlations between variables should be included in the legend of Figure 2 and not in the text.

Line 193. Figure 2 should be self explanatory. The legend must be improved, explaining in it the meaning of the colors used.

Lines 195-206. This sentence does not provide relevant information and should be deleted. In this section, the analytical methodology used must be specified, dispensing with well-known general sentences.

Lines 208-212. Similar reflection as for the previous sentence.

Line 246. What does the value 296.0 represent? AIC or ΔAICc?.

Line 261 (Table 2). If ΔAICc = 296.0, what is the AIC value of the default model?

Line 262. Similarly, what is the AIC value of the optimized model?.

Reviewer 3 Report

Referee’s comments on Bai et al. 2022

’Habitat Suitability Assessment of Black-necked Crane (Grus 2

nigricollis) in the Zoige Grassland Wetland Ecological Function 3

Zone on the Eastern Tibetan Plateau’

The ms addresses a key question in conservation ecology, the development of species-specific conservation strategies based on Species Distribution Modelling. I agree with the choice of Maxent, as the leading software for SDM analyses and the way the authors applied this approach is adequate for the study questions. The study species is an optimal model organism of SDM-based conservation planning, as it is an umbrella, flagship and endangered bird of declining wetlands with a high societal value.

Although the ms is well written, the statements are clearly formulated, the text needs moderate levels of English editing.

I recommend to extend the description of biological relevance of the model specification of Maxent, such as feature combination and regularization multiplier (RM). Importantly, the results show high AUC values.

In Lines 287-290, it is necessary to explain the threshold values for the broader audience with less knowledge in SDM modelling.

The conclusions are in line with the conservation ecology of the Black-necked Cranes.

Round 2

Reviewer 2 Report

Regarding the suggestions made to the first version of the manuscript, the authors have resolved adequately.

Specific Comments

Line 297. Authors must indicate the direction of the effect (positive or negative) of the variables included in the model. This is necessary to properly interpret the results obtained.
